# Investigating the Challenges and Benefits of Engaging in Peer Support via Videoconferencing for People with Spinal Cord Injury

**DOI:** 10.3390/ijerph19084585

**Published:** 2022-04-11

**Authors:** Linda Barclay, Aislinn Lalor

**Affiliations:** 1Department of Occupational Therapy, Monash University, Frankston, VIC 3199, Australia; aislinn.lalor@monash.edu; 2Rehabilitation, Ageing and Independent Living (RAIL) Research Centre, Monash University, Frankston, VIC 3199, Australia

**Keywords:** spinal cord injury, peer support, videoconferencing, lived experience, online support

## Abstract

Background: One of the greatest challenges faced by people following a spinal cord injury is reintegrating into the community. Peer mentors are people who have had shared experiences of disadvantage and distress and have successfully navigated their way through the associated challenges to lead meaningful lives. Historically, peer mentoring services have been predominantly delivered via face-to-face interactions. Little is known about the experience of people with spinal cord injury engaging in online peer support services, and what the challenges and benefits are of this mode of delivery. Methods: An anonymous online survey consisting of closed and open response questions was used to collect data. Quantitative data were analysed descriptively and qualitative data were analysed using inductive content analysis. Results: Positive benefits of engaging in peer support via videoconferencing included convenience and social connectedness. The main barriers were problems with Wi-Fi and internet connections, inconsistencies between platforms and having to learn new platforms. Even though responses were mixed when comparing videoconferencing to face-to-face peer support, most participants felt socially connected. Conclusions: Addressing barriers through the provision of appropriate technology, and targeted and individualised assistance, is important to facilitate uptake of online peer support for people with spinal cord injury.

## 1. Introduction

One of the greatest challenges faced by people following a spinal cord injury (SCI) is reintegrating into the community and finding ways to participate in previously valued roles including paid work, education and social [1,2]. Peer mentors are people who have had shared experiences of disadvantage and distress and have successfully navigated their way through the associated challenges to lead meaningful lives [3]. Due to this shared lived experience, peer mentors are well placed to provide practical, emotional and informational support to others [1,4,5]. There is growing evidence of the efficacy of peer-led interventions for people with SCI [6,7,8].

People with SCI and other disabilities are often socially isolated, have low rates of employment and can be difficult to engage in community activities or support services [9]. Barriers to accessing face-to-face services include living rurally or remotely or being unwell and unable to travel [10]. Within the field of SCI, peer mentoring services and interventions are generally provided outside the formal system of care by not-for-profit community organisations [6]. Historically, these services have been predominantly delivered via face-to-face interactions, individually and in-group settings. There has been some reluctance by organisations to implement online or virtual peer support due to challenges around technology for both the organisations and the service users. The onset of the COVID-19 pandemic exacerbated the social isolation of people with disabilities, due to the reduction in access to face-to-face services and supports due to lockdowns, and anxiety about going out due to fear of contracting the virus [11]. However, one of the positive outcomes of the COVID-19 pandemic has been the rapid shift from face-to-face delivery of services to telehealth and telecare by using videoconferencing (VC), particularly in the medical and rehabilitation field [12].

There are a number of benefits for people with significant disabilities receiving supports and services via VC, including saving time and money, and providing a more convenient way to access support, particularly for those living in rural and remote areas [12,13,14]. However, it not known how likely people with SCI are to engage in this form of peer support, and what the challenges are for accessing support in this way. There can be barriers including lack of expertise and comfort with technology, and limited access to a computer or mobile technology [15,16]. There are examples of research regarding engagement in online peer support in other health areas, such as management of chronic conditions [17], post cancer support [18] and mental health [19]; however, the use of online peer support services by people with SCI is an under-researched area. A recently published study found initial evidence that SCI peer mentors are equally effective at forming positive, supportive relationships with mentees regardless of whether the interaction occurs via telephone or videoconferencing [20]. This provides evidence to support the delivery of peer support via VC, but also highlights the need to obtain additional evidence in this area.

Considering the valuable role that peer support has for people with SCI, it is important to understand whether people with SCI want to engage in peer support in this way. Understanding the challenges and benefits for people engaging in online peer support can inform providers of SCI peer support services about ways to maximise access for those people who are not able to attend or do not want to attend face-to-face peer support. The shift to online delivery of supports resulting from the COVID-19 pandemic provided a unique opportunity to obtain information to inform service delivery in this area. The aim of this research is to identify the challenges and benefits for people with SCI who engage in peer support via VC.

## 2. Materials and Methods

### 2.1. Study Design

This exploratory, descriptive study used an anonymous online survey to gather data. Recruiting people with SCI living in the community is known to be challenging. In addition, recruitment occurred during severe lockdowns due to COVID-19. Therefore, this format was chosen to maximise the number of participants as it was convenient and would only take a short time to complete. Surveys, as with any study design, have potential sources of bias including: (i) question design (e.g., wording, missing data for intended purpose, faulty scale, leading questions, inconsistency); (ii) questionnaire design (e.g., formatting, length, question structure); and administration. To reduce the potential of these biases, we employed a number of strategies including developing the survey based on a comprehensive literature search, in conjunction with the first named author’s expertise and knowledge regarding community integration and peer mentoring in SCI [6], and the second author’s knowledge and expertise in designing surveys [21]. The survey was piloted prior to being released. Participants were first asked demographic questions, followed by forced Likert response, yes/no, and multiple option responses, including a space for “other” information. Questions were worded to make it as easy as possible for participants to understand. For example, when asked about the level or type of injury, rather than using the ASIA Impairment Scale [22], descriptors were provided e.g., Paraplegia–no involvement of upper limbs, require a wheelchair and some assistance with self-care. Open-ended options were placed after three questions to allow participants to elaborate on their answers, with no limits to the length of response. An additional opportunity for any other comments was also provided at the end. Participants were also asked to indicate what VC platforms or programs and other social media they have used (multiple options were available). The survey took approximately 20–30 min to complete. Qualtrics software, Version 2020 [23] was used to publish the survey. The questions asked are available as a Appendix A.

### 2.2. Participants

Participants were recruited via advertisements placed in three Australian-based SCI community organisation newsletters inviting people with SCI, aged over 18 years, who have been receiving peer support from a community organisation via VC (e.g., Zoom, FaceTime, Skype) to complete a short online survey. Posts were also placed on relevant social media sites including LinkedIn, Facebook and Twitter inviting participants. As this was a self-report survey completed by people with SCI who had engaged in peer support via VC, no exclusion criteria applied. The survey was commenced 39 times with 29 surveys fully completed and submitted. The additional incomplete surveys did not provide sufficient data to be useful, and therefore were omitted from the analyses.

### 2.3. Data Collection and Analysis

The survey was open from 19 May 2020–1 July 2021. Completed data were downloaded from the Qualtrics platform into an Excel spreadsheet. Due to the small sample size, numerical data from questions 1–21 were analysed descriptively and reported as count and frequency (*n* (%)) for categorical variables, with associated tables generated. Qualitative data from the open-ended questions were analysed using inductive content analysis [24]. The first author independently coded the text responses line by line, using an inductive process. Themes with illustrative quotes were developed and then discussed with the second author to enhance rigour [25]. A consensus was reached with final themes named and reported here. Further details regarding coding are available from the authors on request.

## 3. Results

### 3.1. Quantitative Data

Twenty-nine participants completed the survey. Table 1 provides an overview of the characteristics of the 29 included participants. The majority of participants were aged between 40–69 years (82.8%), with a relatively even spread of injury levels and types. Time since injury was between 1 to 52 years (median 15 years). Two-thirds of the participants were either living with a partner/spouse (51.7%) or a parent (17.2%), and 13 were receiving assistance from a paid carer (44.8%). According to their postcode, approximately one-third (31.0%) of participants reported living rurally or regionally, although 13.8% did not report their location. More than half (62.1%) of participants reported being employed in some capacity and 65.4% were in receipt of some form of insurance funding. High levels of education were evident in the sample, with 72.4% reporting either a technical college qualification or university degree. Approximately half (51.7%) had an annual income of less than AUD $50,000.

Almost two-thirds of participants (64.3%) had some or a great deal of experience with VC, with over half (55.1%) either somewhat or very comfortable with VC.

There was a high percentage of reported use of a number of VC platforms: Zoom (82.8%), Facebook Messenger (75.9%) and Facetime (65.5%). Participant experience with social media varied, with a high percentage of participants who used the social media platform Facebook (75.9%).

Table 2 outlines the identified challenges and benefits of using VC for peer support. Most participants noted positive benefits, which included being able to socially connect when they otherwise could not (72.4%) and being convenient due to not needing to leave the house (72.4%). Over two-thirds (69%) felt that VC was a successful method for receiving peer support services. The main issues identified in the survey were problems with Wi-Fi or internet connections (37.9%), inconsistencies between platforms (34.5%) and having to learn new platforms (37.9%). Even though there were mixed responses when comparing VC to face-to-face peer support, the majority (75.9%) felt that peer support via VC enabled them to feel socially connected.

### 3.2. Qualitative Data

Three themes identified from the qualitative data collected in the open text response sections of the survey are:The positive aspects of peer support via teleconferencing;The negative aspects of peer support via teleconferencing;A temporary solution during COVID-19.

These themes are discussed below with illustrative quotes from participants.

#### 3.2.1. Positive Aspects of Peer Support via Teleconferencing

One of the most commonly stated positive aspects of teleconferencing was the convenience. A number of participants noted that they could still participate even if they were forced to be at home in bed, were in pain and not able to travel or lived long distances away from the centre. One participant said: “I think the teleconference is a great alternative for people with disabilities”. Others noted that they could save time and money on transport costs, parking and carers. One participant stated: “Videoconferencing is easy and convenient and gives you the ability to always be connected and involved”. Being able to participate in peer support even if in bed or in pain and being able to use the camera off option were benefits. Meeting new people with SCI and hearing about the experiences of other people with SCI was one of the advantages of engaging in peer support this way.

The technology itself allowed possibilities that would not normally exist, as one participant stated, “The technology allows us to share screen, documents and photos”.

Others found it inspiring that many people could be included and that they could be anywhere in the country or even the world. One participant said: “In the past, I believe the Webinars were held in Melbourne making it inaccessible to many. Having the Webinars on Zoom allowed people from all over Victoria to participate-including me”. Another said: “Best thing ever to come out of COVID-19”.

Most participants felt that already knowing the person providing the peer support was an advantage. This was due to there already being an established relationship and feeling comfortable with the person, therefore not needing extra time to establish rapport. One participant said: “I personally like to have met the person, so I know if we will connect”.

#### 3.2.2. Negative Aspects of Peer Support via Teleconferencing

Very few participants articulated negative aspects or challenges. One of the challenges noted was the lack of non-verbal cues. One participant said, “I like to meet to read people and their body language”. Other challenges included distractions at home, not having the correct device or having to share a device, not having a suitable or private environment and occasional bad connections. Some also felt that it was not as personal as face-to-face interactions.

#### 3.2.3. A Temporary Solution during COVID-19

It was commonly noted by participants that VC offered a viable temporary alternative during COVID-19 when they were isolating at home; however, many were keen to get back to face-to-face peer support. Comments included: “Videoconferencing is a good adjunct and better than nothing”, “It’s been a good way to see people in COVID time. (I’m) keen to get back to face-to-face meetings” and “I am really keen to go back to face-to-face”.

## 4. Discussion

In order to inform SCI organisations that provide peer support online, this research aimed to understand the challenges and benefits of using VC to engage in peer support services. As this study only recruited people who had engaged in peer support via VC, it was not possible to gain any understanding of the reasons people did not use VC for peer support.

The majority of our sample had a low income. This is consistent with larger demographic studies of people with SCI [26]. This group also reported the lowest levels of previous experience with VC, compared to those with higher incomes. It is known that those with lower incomes have lower ownership rates of computers and smartphones, and less access to hi-speed internet [27]. Rehabilitation funders should consider the provision of mobile devices and support to access internet services as a high priority to maximise social engagement of their service users.

In comparison to typical SCI samples, in which employment rates are approximately 30–35% [28,29], there was a high number of people employed either full-time, part-time or self-employed (62.1%) in our sample. In addition, over 72% of our participants had either a technical college qualification or a university degree. Therefore, our sample does have a bias, with participants having a higher level of employment and being more highly educated than is typical of most samples of people with SCI [28,30]. It is logical to assume that employed people, and those more highly educated, are likely to have higher levels of use and comfort with technology, therefore taking up the opportunity to use online peer support. People with SCI who are less educated may require training and support to get started with VC and ongoing assistance to trouble shoot should problems that arise. Organisations that provide peer services could arrange peer buddy systems, pairing up confident internet users with those less confident.

The majority of the comments received from participants in this study were positive. Although meeting by VC can change some dynamics in conversations, particularly in relation to receiving limited non-verbal feedback, participants identified that peer support provided this way can facilitate strong connections, which they valued. This is similar to findings from two other programs—one that implemented e-mentoring for youth with physical disabilities preparing for employment, and another that implemented a peer-support programme by group VC for isolated carers of people with dementia [16,31]. For these positive outcomes to occur consistently for people with disabilities, individualised support such as physical assistance setting up and accessing devices should be provided as part of the rehabilitation services [32].

The Australian Bureau of Statistics (2019) found that 15.3% of people in Australia with a physical restriction did not have access to a computer or mobile technology, 23.4% lacked confidence or knowledge in accessing the internet and 14.3% had no access to a computer or mobile technology [15]. The digital divide between those with disabilities and those without is well documented [33,34,35], as is the difficulty that people in rural and remote areas have in accessing the internet [36,37]. Infrastructure barriers, such as poor internet coverage, need to be urgently addressed by relevant jurisdictions to minimise the disadvantage that people with disabilities already experience. In addition, funders of rehabilitation and disability services should consider providing not only appropriate technology for people with disabilities to use online platforms, but also provide training and ongoing assistance to fund recipients to enable them to engage with online peer support and other services. While these new modes of service delivery have the potential to provide greater flexibility and access for people with disabilities, if insufficient attention is given to the design, implementation and policy aspects of these services, there may be an inability to make use of them [33].

Participants in this study commented on the convenience of being able to participate in peer mentoring from the comfort of home, not having to travel and also that it saves time and money. These findings are similar to the results of studies investigating the delivery of peer support via VC for carers of people with dementia [16] and for people experiencing substance abuse recovery [38]. It is well understood that people with significant disability, such as SCI, have challenges travelling long distances and can have limited access to suitable transport which impacts on their ability to participate in the community including getting to groups, appointments or other supports [37]. VC is one way to overcome transport-related barriers [32]. It is recommended that organisations that provide peer support services offer the option of online support to their members. One way of increasing the uptake of online peer support is to target newly injured people during rehabilitation and link them into assistance which can be continued once they return home.

## 5. Conclusions

One of the limitations of this study was the small sample size. Recruitment occurred during severe COVID-19 lockdowns in the state of Victoria and recruitment at this time was particularly challenging. However, our study, albeit with a small sample size, provides initial evidence of the benefits of people with SCI engaging in peer support online. Providing targeted and individualised assistance for less educated or less confident internet users will be important to facilitate uptake in the future. Funders of rehabilitation and disability services should consider providing not only appropriate technology for people with disabilities to use online platforms, but also provide training and ongoing assistance to their recipients to enable them to engage with online peer supports and other services. Depending on jurisdictional responsibility, infrastructure barriers such as poor internet coverage should be addressed to maximise access to online peer support services. Further in-depth exploratory research methods are recommended to obtain a deeper understanding of the experiences of people with SCI utilising this form of peer support.

## Figures and Tables

**Table 1 ijerph-19-04585-t001:** Socio-demographic profile of respondents (*n* = 29).

Variable	Sub-Variable	*n* (%)
Gender	Male	21 (72.4)
	Female	7 (24.1)
	Prefer not to say	1 (3.4)
Age (years)	18–30	1 (3.4)
	30–39	1 (3.4)
	40–49	8 (27.6)
	50–59	8 (27.6)
	60–69	8 (27.6)
	70–79	2 (6.9)
	80 and over	0 (0)
	Not stated	1 (3.4)
Level and type of injury	High quadriplegia/tetraplegia—significant involvement of upper limbs, fully dependent on carers for self-care	2 (6.9)
	Quadriplegia/tetraplegia—involvement of upper limbs, need some/full assistance from carers for self-care	7 (24.1)
	Quadriplegia/tetraplegia—involvement of upper limbs, fully independent in self-care	4 (13.8)
	Paraplegia—no involvement of upper limbs, require a wheelchair and some assistance with self-care	7 (24.1)
	Paraplegia—no involvement of upper limbs, can walk short distances, independent with self-care	3 (10.3)
	Paraplegia—no involvement of upper limbs, able to walk, independent with self-care	2 (6.9)
	Not stated	4 (13.8)
Education (highest level)	Year 10	0 (0)
	Year 11 or 12	5 (17.2)
	TAFE or equivalent	9 (31.0)
	University degree	8 (27.6)
	Higher degree (Masters/doctoral)	4 (13.8)
	Not stated	3 (10.3)
Employment situation ^1^	Employed part-time	12 (41.4)
	Employed full-time	2 (6.9)
	Student	1 (3.4)
	Self-employed/own business	4 (13.8)
	Not seeking employment	2 (6.9)
	Retired	6 (20.7)
	Unemployed seeking work	2 (6.9)
	Unable to work	2 (6.9)
Living situation	Alone	6 (20.7)
	With parents	5 (17.2)
	With partner/spouse	15 (51.7)
	Not stated	3 (10.3)
Housing situation	Own home with no mortgage	10 (34.5)
	Own home with mortgage	6 (20.7)
	Home owned by someone else (e.g., parents)	6 (20.7)
	Renting	4 (13.8)
	Not stated	3 (10.3)
Living location	Urban	16 (55.2)
	Regional	4 (13.8)
	Rural	5 (17.2)
	Not stated	4 (13.8)
Funding ^2^	Transport accident insurance (Australia)	9 (31.0)
	Work insurance (Australia)	1 (3.4)
	Disability funding (NDIS) (Australia)	8 (27.6)
	MyAgedCare (Australia)	1 (3.4)
	Other	4 (13.8)
	Not stated	6 (20.7)
Paid personal carer	Yes	13 (44.8)
	No	12 (41.4)
	Not stated	4 (13.8)
Experience with VC platforms ^1^	Skype	17 (58.6)
	Zoom	24 (82.8)
	Google hangouts	2 (6.9)
	Microsoft teams	11 (37.9)
	Facetime	19 (65.5)
	Facebook messenger	22 (75.9)
	WhatsApp	14 (48.3)
	Viber	10 (34.5)
	Other	2 (6.90
Experience with other social media ^1^	Facebook	22 (75.9)
	Instagram	8 (27.6)
	Twitter	4 (13.8)
	Snapchat	3 (10.3)
	Other	2 (6.9)

^1^ Could choose more than one option. Note. TAFE: Technical and Further Education; NDIS = National Disability Insurance Scheme. ^2^ Transport accident insurance (Australia)—insurance cover for all people involved in road accidents. Work insurance (Australia)—insurance cover for all accidents that occur at work. Disability funding (NDIS) (Australia)—services and support for people 18–65 years with permanent disability. MyAgedCare (Australia)—services and support for people aged over 65 years.

**Table 2 ijerph-19-04585-t002:** Experience, benefits and challenges of using VC for peer support.

Variable	*n* = 29	Percentage (%)
**Previous experience videoconferencing**		
No experience	7	24.1
Some experience	14	48.3
A great deal of experience	5	17.3
Not stated	3	10.3
**Experience using videoconferencing (VC) for peer support**		
Very poor	0	0
Somewhat poor	0	0
Neutral	5	17.2
Somewhat successful	9	31
Very successful	11	37.9
Not stated	4	13.9
**Assistance with set-up and from whom**		
**Yes**	5	17.2
Family member	1	3.4
Service provider/therapist	0	0
Personal care assistant	4	13.8
Other	0	0
**No**	21	72.4
Not stated	3	10.3
**Positive aspects of using videoconferencing for peer support ***		
Enabled me to connect when otherwise I could not	21	72.4
Convenient as I did not have to leave the house	21	72.4
New skill and form of communication I have not tried previously	11	37.9
Convenient as I could pick the time	12	41.4
Other	4	13.8
**Challenges of using videoconferencing for peer support ***		
Wi-fi/internet connection issues	11	37.9
Needing help to get set-up	3	10.3
Cost	2	6.9
Learning new platform/program	11	37.9
Inconsistencies between platforms	10	34.5
Not getting non-verbal feedback from another person	6	20.7
Other	4	13.8
**Comparison to face-to-face**		
Compared very poorly	0	0
Compared somewhat poorly	6	20.7
Neutral	8	27.6
Compared somewhat well	8	27.6
Compared very well	4	13.8
Not stated	3	10.3
**Videoconferencing helped me to feel socially connected**		
Yes	22	75.9
No	2	6.9
Not stated	5	17.2
**Person providing peer support**		
Someone new	6	20.7
Someone known	16	55.2
Both someone new and someone known	3	10.3
Not stated	4	13.8

* Could choose more than one option.

## Data Availability

The data presented in this study are available on request from the corresponding author. The data are not publicly available due to privacy and ethical restrictions.

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
