# Peer review of "Investigating the Challenges and Benefits of Engaging in Peer Support via Videoconferencing for People with Spinal Cord Injury"

_ijerph, 2022, doi:10.3390/ijerph19084585_

Round 1

Reviewer 1 Report

Thank you for the opportunity to review this paper reporting on a study of video conferencing (VC) for peer support for people living with spinal cord injury (SCI) during the initial stages of Covid.

This is a small survey study of the experiences of people with SCI who used VC during Covid to maintain social contact with peer support. The survey employed fairly closed questions to examine the participant's experiences, and it is a little unfortunate the the lived experience of this approach to peer support wasn't examined in a broader way. Instead the authors chose to ask about positive experiences and challenges using this form of peer support, and along with the small sample size offers a snapshot of the experiences.

It is useful to understand who responded (and perhaps why these people chose to do so), and I would have liked to have known more about the recruitment strategy - do the organisations represent the majority of people with SCI? Was this the best way to recruit participants - given that so many of them linked in via Facebook, would social media recruitment have been more successful at obtaining respondents? I was curious as to why so many didn't complete the survey, and it would be helpful if the authors could address the limitations of this study and what they would recommend be done in future work.

Did the authors consider running a focus group? or augmenting their study with interviews to obtain a deeper understanding of the experiences people with SCI had? I'm left with a sense that there is much more that could have been understood had these approaches been incorporated, because while the recommendations are quite clear, there are some assumptions about how valuable participants found using this approach. Asking them how they might have been better supported might offer some alternative solutions to the issues raised in the paper.

For international readers, the call for government support to enable better access might not be entirely credible - might this be dependent on the way the local jurisdiction works? eg in the US it's unlikely to be a Federal Government concern.

Overall, though, I think the simple aims of this study were met, and the reporting is clear and accurate.

Reviewer 2 Report

Dear authors,

I am glad to have the opportunity to review this interesting manuscript, which it is well written and the text is very understandable and organized.

Specific comments:

- Participants. Give the exclusion criteria.

- Variables. Items of surveys. Define the potential sources of bias.

- It is know that non-verbal and verbal communication skills are very important in the medical and rehabilitation field. So, how can researchers or clinicians control these skills through the use of videoconferencing tool?

Round 2

Reviewer 2 Report

Dear authors,

I appreciate the possibility to review the revised version of this manuscript. The authors have improved the quality of the manuscript with your corrections.